# Implicit Semantic Data Augmentation for Deep Networks

**Yulin Wang**[1][*]  **Xuran Pan**[1][*]  **Shiji Song**[1]  **Hong Zhang**[2]  **Cheng Wu**[1]  **Gao Huang**[1][†]

[1]Department of Automation, Tsinghua University, Beijing, China
Beijing National Research Center for Information Science and Technology (BNRist),
[2]Baidu Inc., China
{yulin.bh, fykalviny}@gmail.com, pxr18@mails.tsinghua.edu.cn,
{shijis, wuc, gaohuang}@tsinghua.edu.cn

## Abstract

In this paper, we propose a novel *implicit semantic data augmentation (ISDA)* approach to complement traditional augmentation techniques like flipping, translation or rotation. Our work is motivated by the intriguing property that deep networks are surprisingly good at linearizing features, such that certain directions in the deep feature space correspond to meaningful semantic transformations, e.g., adding sunglasses or changing backgrounds. As a consequence, translating training samples along many semantic directions in the feature space can effectively augment the dataset to improve generalization. To implement this idea effectively and efficiently, we first perform an online estimate of the covariance matrix of deep features for each class, which captures the intra-class semantic variations. Then random vectors are drawn from a zero-mean normal distribution with the estimated covariance to augment the training data in that class. Importantly, instead of augmenting the samples explicitly, we can directly minimize an upper bound of the *expected* cross-entropy (CE) loss on the augmented training set, leading to a highly efficient algorithm. In fact, we show that the proposed ISDA amounts to minimizing a novel robust CE loss, which adds negligible extra computational cost to a normal training procedure. Although being simple, ISDA consistently improves the generalization performance of popular deep models (ResNets and DenseNets) on a variety of datasets, e.g., CIFAR-10, CIFAR-100 and ImageNet. Code for reproducing our results is available at *https://github.com/blackfeather-wang/ISDA-for-Deep-Networks*.

## 1  Introduction

Data augmentation is an effective technique to alleviate the overfitting problem in training deep networks [1, 2, 3, 4, 5]. In the context of image recognition, this usually corresponds to applying content preserving transformations, e.g., cropping, horizontal mirroring, rotation and color jittering, on the input samples. Although being effective, these augmentation techniques are not capable of performing semantic transformations, such as changing the background of an object or the texture of a foreground object. Recent work has shown that data augmentation can be more powerful if (class identity preserving) semantic transformations are allowed [6, 7, 8]. For example, by training a generative adversarial network (GAN) for each class in the training set, one could then sample an infinite number of samples from the generator. Unfortunately, this procedure is computationally intensive because training generative models and inferring them to obtain augmented samples are

---

[*]Equal contribution.
[†]Corresponding author.

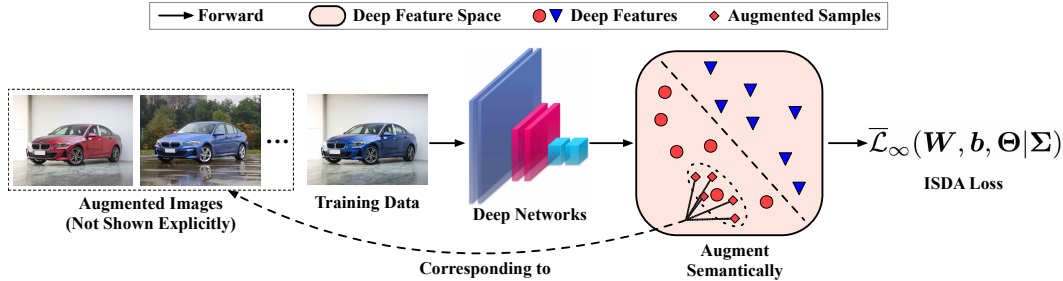

Figure 1: An overview of ISDA. Inspired by the observation that certain directions in the feature space correspond to meaningful semantic transformations, we augment the training data semantically by translating their features along these semantic directions, without involving auxiliary deep networks. The directions are obtained by sampling random vectors from a zero-mean normal distribution with dynamically estimated class-conditional covariance matrices. In addition, instead of performing augmentation explicitly, ISDA boils down to minimizing a closed-form upper-bound of the expected cross-entropy loss on the augmented training set, which makes our method highly efficient.

both nontrivial tasks. Moreover, due to the extra augmented data, the training procedure is also likely to be prolonged.

In this paper, we propose an implicit semantic data augmentation (ISDA) algorithm for training deep image recognition networks. The ISDA is highly efficient as it does not require training/inferring auxiliary networks or explicitly generating extra training samples. Our approach is motivated by the intriguing observation made by recent work showing that the features deep in a network are usually linearized [9, 10]. Specifically, there exist many semantic directions in the deep feature space, such that translating a data sample in the feature space along one of these directions results in a feature representation corresponding to another sample with the same class identity but different semantics. For example, a certain direction corresponds to the semantic translation of "make-bespectacled". When the feature of a person, who does not wear glasses, is translated along this direction, the new feature may correspond to the same person but with glasses (The new image can be explicitly reconstructed using proper algorithms as shown in [9]). Therefore, by searching for many such semantic directions, we can effectively augment the training set in a way complementary to traditional data augmenting techniques.

However, explicitly finding semantic directions is not a trivial task, which usually requires extensive human annotations [9]. In contrast, sampling directions randomly is efficient but may result in meaningless transformations. For example, it makes no sense to apply the "make-bespectacled" transformation to the "car" class. In this paper, we adopt a simple method that achieves a good balance between effectiveness and efficiency. In specific, we perform an online estimate of the covariance matrix of the features for *each* class, which captures the intra-class variations. Then we sample directions from a zero-mean multi-variate normal distribution with the estimated covariance, and apply them to the features of training samples in that class to augment the dataset. In this way, the chance of generating meaningless semantic transformations can be significantly reduced.

To further improve the efficiency, we derive a closed-form upper bound of the *expected* cross-entropy (CE) loss with the proposed data augmentation scheme. Therefore, instead of performing the augmentation procedure explicitly, we can directly minimize the upper bound, which is, in fact, a novel robust loss function. As there is no need to generate explicit data samples, we call our algorithm *implicit semantic data augmentation (ISDA)*. Compared to existing semantic data augmentation algorithms, the proposed ISDA can be conveniently implemented on top of most deep models without introducing auxiliary models or noticeable extra computational cost.

Although being simple, the proposed ISDA algorithm is surprisingly effective, and complements existing non-semantic data augmentation techniques quite well. Extensive empirical evaluations on several competitive image classification benchmarks show that ISDA consistently improves the generalization performance of popular deep networks, especially with little training data and powerful traditional augmentation techniques.

## 2 Related Work

In this section, we briefly review existing research on related topics.

**Data augmentation** is a widely used technique to alleviate overfitting in training deep networks. For example, in image recognition tasks, data augmentation techniques like random flipping, mirroring and rotation are applied to enforce certain invariance in convolutional networks [4, 5, 3, 11]. Recently, automatic data augmentation techniques, e.g., AutoAugment [12], are proposed to search for a better augmentation strategy among a large pool of candidates. Similar to our method, learning with marginalized corrupted features [13] can be viewed as an implicit data augmentation technique, but it is limited to simple linear models. Complementarily, recent research shows that semantic data augmentation techniques which apply class identity preserving transformations (e.g. changing backgrounds of objects or varying visual angles) to the training data are effective as well [14, 15, 6, 8]. This is usually achieved by generating extra semantically transformed training samples with specialized deep structures such as DAGAN [8], domain adaptation networks [15] or other GAN-based generators [14, 6]. Although being effective, these approaches are nontrivial to implement and computationally expensive, due to the need to train generative models beforehand and infer them during training.

**Robust loss function.** As shown in the paper, ISDA amounts to minimizing a novel robust loss function. Therefore, we give a brief review of related work on this topic. Recently, several robust loss functions are proposed for deep learning. For example, the $L_q$ loss [16] is a balanced noise-robust form for the cross entropy (CE) loss and mean absolute error (MAE) loss, derived from the negative Box-Cox transformation. Focal loss [17] attaches high weights to a sparse set of hard examples to prevent the vast number of easy samples from dominating the training of the network. The idea of introducing large margin for CE loss has been proposed in [18, 19, 20]. In [21], the CE loss and the contrastive loss are combined to learn more discriminative features. From a similar perspective, center loss [22] simultaneously learns a center for deep features of each class and penalizes the distances between the samples and their corresponding class centers in the feature space, enhancing the intra-class compactness and inter-class separability.

**Semantic transformations in deep feature space.** Our work is motivated by the fact that high-level representations learned by deep convolutional networks can potentially capture abstractions with semantics [23, 10]. In fact, translating deep features along certain directions is shown to be corresponding to performing meaningful semantic transformations on the input images. For example, deep feature interpolation [9] leverages simple interpolations of deep features from pre-trained neural networks to achieve semantic image transformations. Variational Autoencoder(VAE) and Generative Adversarial Network(GAN) based methods [24, 25, 26] establish a latent representation corresponding to the abstractions of images, which can be manipulated to edit the semantics of images. Generally, these methods reveal that certain directions in the deep feature space correspond to meaningful semantic transformations, and can be leveraged to perform semantic data augmentation.

## 3 Method

Deep networks are known to excel at forming high-level representations in the deep feature space [4, 5, 9, 27], where the semantic relations between samples can be captured by the relative positions of their features [10]. Previous work has demonstrated that translating features towards specific directions corresponds to meaningful semantic transformations when the features are mapped to the input space [9, 28, 10]. Based on this observation, we propose to directly augment the training data in the feature space, and integrate this procedure into the training of deep models.

The proposed implicit semantic data augmentation (ISDA) has two important components, i.e., online estimation of class-conditional covariance matrices and optimization with a robust loss function. The first component aims to find a distribution from which we can sample meaningful semantic transformation directions for data augmentation, while the second saves us from explicitly generating a large amount of extra training data, leading to remarkable efficiency compared to existing data augmentation techniques.

### 3.1 Semantic Transformations in Deep Feature Space

As aforementioned, certain directions in the deep feature space correspond to meaningful semantic transformations like "make-bespectacled" or 'change-view-angle'. This motivates us to augment the training set by applying such semantic transformations on deep features. However, manually searching for semantic directions is infeasible for large scale problems. To address this problem, we propose to approximate the procedure by sampling random vectors from a normal distribution with zero mean and a covariance that is proportional to the intra-class covariance matrix, which captures the variance of samples in that class and is thus likely to contain rich semantic information.

Intuitively, features for the *person* class may vary along the "wear-glasses" direction, while having nearly zero variance along the "has-propeller" direction which only occurs for other classes like the *plane* class. We hope that directions corresponding to meaningful transformations for each class are well represented by the principal components of the covariance matrix of that class.

Consider training a deep network $G$ with weights $\boldsymbol{\Theta}$ on a training set $\mathcal{D} = \{(\boldsymbol{x}_i, y_i)\}_{i=1}^N$, where $y_i \in \{1, \dots, C\}$ is the label of the $i$-th sample $\boldsymbol{x}_i$ over $C$ classes. Let the $A$-dimensional vector $\boldsymbol{a}_i = [a_{i1}, \dots, a_{iA}]^T = G(\boldsymbol{x}_i, \boldsymbol{\Theta})$ denote the deep features of $\boldsymbol{x}_i$ learned by $G$, and $a_{ij}$ indicate the $j$th element of $\boldsymbol{a}_i$.

To obtain semantic directions to augment $\boldsymbol{a}_i$, we randomly sample vectors from a zero-mean multivariate normal distribution $\mathcal{N}(0, \Sigma_{y_i})$, where $\Sigma_{y_i}$ is the class-conditional covariance matrix estimated from the features of all the samples in class $y_i$. In implementation, the covariance matrix is computed in an online fashion by aggregating statistics from all mini-batches. The online estimation algorithm is given in Section A in the supplementary.

During training, $C$ covariance matrices are computed, one for each class. The augmented feature $\tilde{\boldsymbol{a}}_i$ is obtained by translating $\boldsymbol{a}_i$ along a random direction sampled from $\mathcal{N}(0, \lambda\Sigma_{y_i})$. Equivalently, we have

$$\tilde{\boldsymbol{a}}_i \sim \mathcal{N}(\boldsymbol{a}_i, \lambda\Sigma_{y_i}), \tag{1}$$

where $\lambda$ is a positive coefficient to control the strength of semantic data augmentation. As the covariances are computed dynamically during training, the estimation in the first few epochs are not quite informative when the network is not well trained. To address this issue, we let $\lambda = (t/T) \times \lambda_0$ be a function of the current iteration $t$, thus to reduce the impact of the estimated covariances on our algorithm early in the training stage.

## 3.2 Implicit Semantic Data Augmentation (ISDA)

A naive method to implement ISDA is to explicitly augment each $\boldsymbol{a}_i$ for $M$ times, forming an augmented feature set $\{(\boldsymbol{a}_i^1, y_i), \dots, (\boldsymbol{a}_i^M, y_i)\}_{i=1}^N$ of size $MN$, where $\boldsymbol{a}_i^k$ is $k$-th copy of augmented features for sample $\boldsymbol{x}_i$. Then the networks are trained by minimizing the cross-entropy (CE) loss:

$$\mathcal{L}_M(\boldsymbol{W}, \boldsymbol{b}, \boldsymbol{\Theta}) = \frac{1}{N}\sum_{i=1}^N \frac{1}{M}\sum_{k=1}^M -log(\frac{e^{\boldsymbol{w}_{y_i}^T \boldsymbol{a}_i^k + b_{y_i}}}{\sum_{j=1}^C e^{\boldsymbol{w}_j^T \boldsymbol{a}_i^k + b_j}}), \tag{2}$$

where $\boldsymbol{W} = [\boldsymbol{w}_1, \dots, \boldsymbol{w}_C]^T \in \mathcal{R}^{C \times A}$ and $\boldsymbol{b} = [b_1, \dots, b_C]^T \in \mathcal{R}^C$ are the weight matrix and biases corresponding to the final fully connected layer, respectively.

Obviously, the naive implementation is computationally inefficient when $M$ is large, as the feature set is enlarged by $M$ times. In the following, we consider the case that $M$ grows to infinity, and find that an easy-to-compute upper bound can be derived for the loss function, leading to a highly efficient implementation.

**Upper bound of the loss function.** In the case $M \to \infty$, we are in fact considering the expectation of the CE loss under all possible augmented features. Specifically, $\mathcal{L}_\infty$ is given by:

$$\mathcal{L}_\infty(\boldsymbol{W}, \boldsymbol{b}, \boldsymbol{\Theta}|\boldsymbol{\Sigma}) = \frac{1}{N}\sum_{i=1}^N \mathrm{E}_{\tilde{\boldsymbol{a}}_i}[-log(\frac{e^{\boldsymbol{w}_{y_i}^T \tilde{\boldsymbol{a}}_i + b_{y_i}}}{\sum_{j=1}^C e^{\boldsymbol{w}_j^T \tilde{\boldsymbol{a}}_i + b_j}})]. \tag{3}$$

If $\mathcal{L}_\infty$ can be computed efficiently, then we can directly minimize it without explicitly sampling augmented features. However, Eq. (3) is difficult to compute in its exact form. Alternatively, we find that it is possible to derive an easy-to-compute upper bound for $\mathcal{L}_\infty$, as given by the following proposition.

**Proposition 1.** *Suppose that* $\tilde{\boldsymbol{a}}_i \sim \mathcal{N}(\boldsymbol{a}_i, \lambda\Sigma_{y_i})$, *then we have an upper bound of* $\mathcal{L}_\infty$, *given by*

$$\mathcal{L}_\infty(\boldsymbol{W}, \boldsymbol{b}, \boldsymbol{\Theta}|\boldsymbol{\Sigma}) \le \frac{1}{N}\sum_{i=1}^N -log(\frac{e^{\boldsymbol{w}_{y_i}^T \boldsymbol{a}_i + b_{y_i}}}{\sum_{j=1}^C e^{\boldsymbol{w}_j^T \boldsymbol{a}_i + b_j + \frac{\lambda}{2}(\boldsymbol{w}_j^T - \boldsymbol{w}_{y_i}^T)\Sigma_{y_i}(\boldsymbol{w}_j - \boldsymbol{w}_{y_i})}}) \triangleq \overline{\mathcal{L}}_\infty. \tag{4}$$

***Proof.*** According to the definition of $\mathcal{L}_\infty$ in (3), we have:

$$\mathcal{L}_\infty(\boldsymbol{W}, \boldsymbol{b}, \boldsymbol{\Theta}|\boldsymbol{\Sigma}) = \frac{1}{N}\sum_{i=1}^N \mathrm{E}_{\tilde{\boldsymbol{a}}_i}[log(\sum_{j=1}^C e^{(\boldsymbol{w}_j^T - \boldsymbol{w}_{y_i}^T)\tilde{\boldsymbol{a}}_i + (b_j - b_{y_i})})] \tag{5}$$

$$\leq \frac{1}{N}\sum_{i=1}^{N} log(\sum_{j=1}^{C} \mathrm{E}_{\tilde{\boldsymbol{a}}_i}[e^{(\boldsymbol{w}_j^T - \boldsymbol{w}_{y_i}^T)\tilde{\boldsymbol{a}}_i + (b_j - b_{y_i})}]) \tag{6}$$

$$= \frac{1}{N}\sum_{i=1}^{N} log(\sum_{j=1}^{C} e^{(\boldsymbol{w}_j^T - \boldsymbol{w}_{y_i}^T)\boldsymbol{a}_i + (b_j - b_{y_i}) + \frac{\lambda}{2}(\boldsymbol{w}_j^T - \boldsymbol{w}_{y_i}^T)\Sigma_{y_i}(\boldsymbol{w}_j - \boldsymbol{w}_{y_i})}) \tag{7}$$

$$= \overline{\mathcal{L}}_\infty. \tag{8}$$

In the above, the Inequality (6) follows from the Jensen's inequality $\mathrm{E}[logX] \leq log\mathrm{E}[X]$, as the logarithmic function $log(\cdot)$ is concave. The Eq. (7) is obtained by leveraging the moment-generating function:

$$\mathrm{E}[e^{tX}] = e^{t\mu + \frac{1}{2}\sigma^2 t^2}, \quad X \sim \mathcal{N}(\mu, \sigma^2),$$

due to the fact that $(\boldsymbol{w}_j^T - \boldsymbol{w}_{y_i}^T)\tilde{\boldsymbol{a}}_i + (b_j - b_{y_i})$ is a Gaussian random variable, i.e.,

$$(\boldsymbol{w}_j^T - \boldsymbol{w}_{y_i}^T)\tilde{\boldsymbol{a}}_i + (b_j - b_{y_i}) \sim \mathcal{N}\left((\boldsymbol{w}_j^T - \boldsymbol{w}_{y_i}^T)\boldsymbol{a}_i + (b_j - b_{y_i}), \lambda(\boldsymbol{w}_j^T - \boldsymbol{w}_{y_i}^T)\Sigma_{y_i}(\boldsymbol{w}_j - \boldsymbol{w}_{y_i})\right). \quad \square$$

Essentially, Proposition 1 provides a surrogate loss for our implicit data augmentation algorithm. Instead of minimizing the exact loss function $\mathcal{L}_\infty$, we can optimize its upper bound $\overline{\mathcal{L}}_\infty$ in a much more efficient way. Therefore, the proposed ISDA boils down to a novel robust loss function, which can be easily adopted by most deep models. In addition, we can observe that when $\lambda \to 0$, which means no features are augmented, $\overline{\mathcal{L}}_\infty$ reduces to the standard CE loss.

In summary, the proposed ISDA can be simply plugged into deep networks as a robust loss function, and efficiently optimized with the stochastic gradient descent (SGD) algorithm. We present the pseudo code of ISDA in Algorithm 1. Details of estimating covariance matrices and computing gradients are presented in Appendix A.

---

**Algorithm 1** The ISDA Algorithm.

1: **Input:** $\mathcal{D}$, $\lambda_0$
2: Randomly initialize $\boldsymbol{W}$, $\boldsymbol{b}$ and $\boldsymbol{\Theta}$
3: **for** $t = 0$ **to** $T$ **do**
4:      Sample a mini-batch $\{\boldsymbol{x}_i, y_i\}_{i=1}^B$ from $\mathcal{D}$
5:      Compute $\boldsymbol{a}_i = G(\boldsymbol{x}_i, \boldsymbol{\Theta})$
6:      Estimate the covariance matrices $\Sigma_1, \Sigma_2$, $..., \Sigma_C$
7:      Compute $\overline{\mathcal{L}}_\infty$ according to Eq. (4)
8:      Update $\boldsymbol{W}$, $\boldsymbol{b}$, $\boldsymbol{\Theta}$ with SGD
9: **end for**
10: **Output:** $\boldsymbol{W}$, $\boldsymbol{b}$ and $\boldsymbol{\Theta}$

---

## 4 Experiments

In this section, we empirically validate the proposed algorithm on several widely used image classification benchmarks, i.e., CIFAR-10, CIFAR-100 [1] and ImageNet[29]. We first evaluate the effectiveness of ISDA with different deep network architectures on these datasets. Second, we apply several recent proposed non-semantic image augmentation methods in addition to the standard baseline augmentation, and investigate the performance of ISDA. Third, we present comparisons with state-of-the-art robust lost functions and generator-based semantic data augmentation algorithms. Finally, ablation studies are conducted to examine the effectiveness of each component. We also visualize the augmented samples in the original input space with the aid of a generative network.

### 4.1 Datasets and Baselines

**Datasets.** We use three image recognition benchmarks in the experiments. (1) The two CIFAR datasets consist of 32x32 colored natural images in 10 classes for CIFAR-10 and 100 classes for CIFAR-100, with 50,000 images for training and 10,000 images for testing, respectively. In our experiments, we hold out 5000 images from the training set as the validation set to search for the hyper-parameter $\lambda_0$. These samples are also used for training after an optimal $\lambda_0$ is selected, and the results on the test set are reported. Images are normalized with channel means and standard deviations for pre-procession. For the non-semantic data augmentation of the training set, we follow the standard operation in [30]: 4 pixels are padded at each side of the image, followed by a random 32x32 cropping combined with random horizontal flipping. (2) ImageNet is a 1,000-class dataset from ILSVRC2012[29], providing 1.2 million images for training and 50,000 images for validation. We adopt the same augmentation configurations in [2, 4, 5].

**Non-semantic augmentation techniques.** To study the complementary effects of ISDA to traditional data augmentation methods, two state-of-the-art non-semantic augmentation techniques are applied, with and without ISDA. (1) *Cutout* [31] randomly masks out square regions of input during training to regularize the model. (2) *AutoAugment* [32] automatically searches for the best augmentation policies to yield the highest validation accuracy on a target dataset. All hyper-parameters are the same as reported in the papers introducing them.

Table 1: Evaluation of ISDA on CIFAR with different models. The average test error over the last 10 epochs is calculated in each experiment, and we report mean values and standard deviations in three independent experiments. The best results are **bold-faced**.

| Method | Params | CIFAR-10 | CIFAR-100 |
|---|---|---|---|
| ResNet-32 [4] | 0.5M | $7.39 \pm 0.10\%$ | $31.20 \pm 0.41\%$ |
| ResNet-32 + ISDA | 0.5M | $\mathbf{7.09 \pm 0.12\%}$ | $\mathbf{30.27 \pm 0.34\%}$ |
| ResNet-110 [4] | 1.7M | $6.76 \pm 0.34\%$ | $28.67 \pm 0.44\%$ |
| ResNet-110 + ISDA | 1.7M | $\mathbf{6.33 \pm 0.19\%}$ | $\mathbf{27.57 \pm 0.46\%}$ |
| SE-ResNet-110 [33] | 1.7M | $6.14 \pm 0.17\%$ | $27.30 \pm 0.03\%$ |
| SE-ResNet-110 + ISDA | 1.7M | $\mathbf{5.96 \pm 0.21\%}$ | $\mathbf{26.63 \pm 0.21\%}$ |
| Wide-ResNet-16-8 [34] | 11.0M | $4.25 \pm 0.18\%$ | $20.24 \pm 0.27\%$ |
| Wide-ResNet-16-8 + ISDA | 11.0M | $\mathbf{4.04 \pm 0.29\%}$ | $\mathbf{19.91 \pm 0.21\%}$ |
| Wide-ResNet-28-10 [34] | 36.5M | $3.82 \pm 0.15\%$ | $18.53 \pm 0.07\%$ |
| Wide-ResNet-28-10 + ISDA | 36.5M | $\mathbf{3.58 \pm 0.15\%}$ | $\mathbf{17.98 \pm 0.15\%}$ |
| ResNeXt-29, 8x64d [35] | 34.4M | $3.86 \pm 0.14\%$ | $18.16 \pm 0.13\%$ |
| ResNeXt-29, 8x64d + ISDA | 34.4M | $\mathbf{3.67 \pm 0.12\%}$ | $\mathbf{17.43 \pm 0.25\%}$ |
| DenseNet-BC-100-12 [5] | 0.8M | $4.90 \pm 0.08\%$ | $22.61 \pm 0.10\%$ |
| DenseNet-BC-100-12 + ISDA | 0.8M | $\mathbf{4.54 \pm 0.07\%}$ | $\mathbf{22.10 \pm 0.34\%}$ |
| DenseNet-BC-190-40 [5] | 25.6M | $3.52\%$ | $17.74\%$ |
| DenseNet-BC-190-40 + ISDA | 25.6M | $\mathbf{3.24\%}$ | $\mathbf{17.42\%}$ |

Table 2: Evaluation of ISDA with state-of-the-art *non-semantic* augmentation techniques. 'AA' refers to AutoAugment [32]. We report mean values and standard deviations in three independent experiments. The best results are **bold-faced**.

| Dataset | Networks | Cutout [31] | Cutout + ISDA | AA [32] | AA + ISDA |
|---|---|---|---|---|---|
| CIFAR-10 | Wide-ResNet-28-10 [34] | $2.99 \pm 0.06\%$ | $\mathbf{2.83 \pm 0.04\%}$ | $2.65 \pm 0.07\%$ | $\mathbf{2.56 \pm 0.01\%}$ |
| | Shake-Shake (26, 2x32d) [36] | $3.16 \pm 0.09\%$ | $\mathbf{2.93 \pm 0.03\%}$ | $2.89 \pm 0.09\%$ | $\mathbf{2.68 \pm 0.12\%}$ |
| | Shake-Shake (26, 2x112d) [36] | $2.36\%$ | $\mathbf{2.25\%}$ | $2.01\%$ | $\mathbf{1.82\%}$ |
| CIFAR-100 | Wide-ResNet-28-10 [34] | $18.05 \pm 0.25\%$ | $\mathbf{17.02 \pm 0.11\%}$ | $16.60 \pm 0.40\%$ | $\mathbf{15.62 \pm 0.32\%}$ |
| | Shake-Shake (26, 2x32d) [36] | $18.92 \pm 0.21\%$ | $\mathbf{18.17 \pm 0.08\ \%}$ | $17.50 \pm 0.19\%$ | $\mathbf{17.21 \pm 0.33\%}$ |
| | Shake-Shake (26, 2x112d) [36] | $17.34 \pm 0.28\%$ | $\mathbf{16.24 \pm 0.20\ \%}$ | $15.21 \pm 0.20\%$ | $\mathbf{13.87 \pm 0.26\%}$ |

**Baselines.** Our method is compared to several baselines including state-of-the-art robust loss functions and generator-based semantic data augmentation methods. (1) *Dropout* [37] is a widely used regularization approach which randomly mutes some neurons during training. (2) *Large-margin softmax loss* [18] introduces large decision margin, measured by a cosine distance, to the standard CE loss. (3) *Disturb label* [38] is a regularization mechanism that randomly replaces a fraction of labels with incorrect ones in each iteration. (4) *Focal loss* [17] focuses on a sparse set of hard examples to prevent easy samples from dominating the training procedure. (5) *Center loss* [22] simultaneously learns a center of features for each class and minimizes the distances between the deep features and their corresponding class centers. (6) $L_q$ *loss* [16] is a noise-robust loss function, using the negative Box-Cox transformation. (7) For generator-based semantic augmentation methods, we train several state-of-the-art GANs [39, 40, 41, 42], which are then used to generate extra training samples for data augmentation. For fair comparison, all methods are implemented with the same training configurations when it is possible. Details for hyper-parameter settings are presented in Appendix B.

**Training details.** For deep networks, we implement the ResNet, SE-ResNet, Wide-ResNet, ResNeXt, DenseNet and PyramidNet on CIFAR, and ResNet, ResNeXt on ImageNet. Detailed configurations for these models are given in Appendix B. The hyper-parameter $\lambda_0$ for ISDA is selected from the set $\{0.1, 0.25, 0.5, 0.75, 1\}$ according to the performance on the validation set. On ImageNet, due to GPU memory limitation, we approximate the covariance matrices by their diagonals, i.e., the variance of each dimension of the features. The best hyper-parameter $\lambda_0$ is selected from $\{1, 2.5, 5, 7.5, 10\}$.

### 4.2 Main Results

Table 1 presents the performance of several state-of-the-art deep networks with and without ISDA. It can be observed that ISDA consistently improves the generalization performance of these models, especially with fewer training samples per class. On CIFAR-100, for relatively small models like ResNet-32 and ResNet-110, ISDA reduces test errors by about $1\%$, while for larger models like

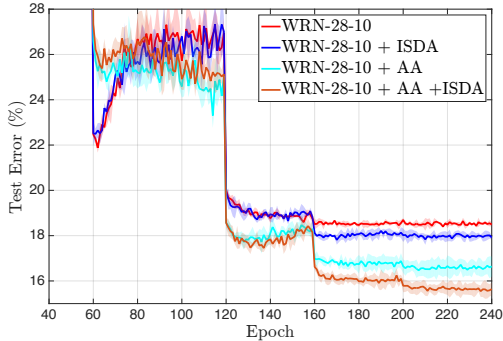

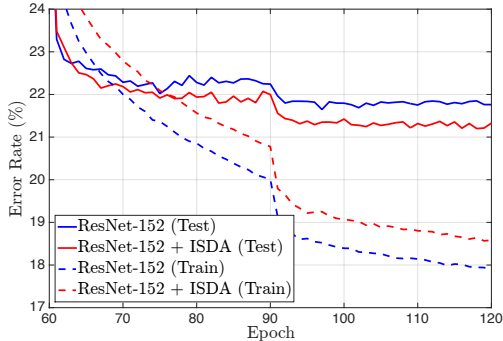

Figure 2: Curves of test errors on CIFAR-100 with Wide-ResNet (WRN).

Figure 3: Training and test errors on ImageNet.

Table 3: Comparisons with the state-of-the-art methods. We report mean values and standard deviations of the test error in three independent experiments. Best results are **bold-faced**.

| Method | ResNet-110 | | Wide-ResNet-28-10 | |
|---|---|---|---|---|
| | CIFAR-10 | CIFAR-100 | CIFAR-10 | CIFAR-100 |
| Large Margin [18] | 6.46±0.20% | 28.00±0.09% | 3.69±0.10% | 18.48±0.05% |
| Disturb Label [38] | 6.61±0.04% | 28.46±0.32% | 3.91±0.10% | 18.56±0.22% |
| Focal Loss [17] | 6.68±0.22% | 28.28±0.32% | 3.62±0.07% | 18.22±0.08% |
| Center Loss [22] | 6.38±0.20% | 27.85±0.10% | 3.76±0.05% | 18.50±0.25% |
| $L_q$ Loss [16] | 6.69±0.07% | 28.78±0.35% | 3.78±0.08% | 18.43±0.37% |
| WGAN [39] | 6.63±0.23% | - | 3.81±0.08% | - |
| CGAN [40] | 6.56±0.14% | 28.25±0.36% | 3.84±0.07% | 18.79±0.08% |
| ACGAN [41] | 6.32±0.12% | 28.48±0.44% | 3.81±0.11% | 18.54±0.05% |
| infoGAN [42] | 6.59±0.12% | 27.64±0.14% | 3.81±0.05% | 18.44±0.10% |
| Basic | 6.76±0.34% | 28.67±0.44% | - | - |
| Basic + Dropout | 6.23±0.11% | 27.11±0.06% | 3.82±0.15% | 18.53±0.07% |
| ISDA | 6.33±0.19% | 27.57±0.46% | - | - |
| ISDA + Dropout | **5.98±0.20%** | **26.35±0.30%** | **3.58±0.15%** | **17.98±0.15%** |

Wide-ResNet-28-10 and ResNeXt-29, 8x64d, our method outperforms the competitive baselines by nearly 0.7%. Compared to ResNets, DenseNets generally suffer less from overfitting due to their architecture design, thus appear to benefit less from our algorithm.

Table 2 shows experimental results with recent proposed powerful traditional image augmentation methods (i.e. Cutout [31] and AutoAugment [32]). Interestingly, ISDA seems to be even more effective when these techniques exist. For example, when applying AutoAugment, ISDA achieves performance gains of 1.34% and 0.98% on CIFAR-100 with the Shake-Shake (26, 2x112d) and the Wide-ResNet-28-10, respectively. Notice that these improvements are more significant than the standard situations. A plausible explanation for this phenomenon is that non-semantic augmentation methods help to learn a better feature representation, which makes semantic transformations in the deep feature space more reliable. The curves of test errors during training on CIFAR-100 with Wide-ResNet-28-10 are presented in Figure 2. It is clear that ISDA achieves a significant improvement after the third learning rate drop, and shows even better performance after the fourth drop.

Table 4 presents the performance of ISDA on the large scale ImageNet dataset. It can be observed that ISDA reduces Top-1 error rate by 0.54% for the ResNeXt-50 model. The training and test error curves of ResNet-152 are shown in Figure 3. Notably, ISDA achieves a slightly higher training error but a lower test error, indicating that ISDA performs effective regularization on deep networks.

Table 4: Evaluation of ISDA on ImageNet.

| Method | Top-1 | Top-5 |
|---|---|---|
| ResNet-50 [4] | 23.58% | 6.92% |
| ResNet-50 + ISDA | **23.30%** | **6.82%** |
| ResNet-152 [4] | 21.65% | 6.01% |
| ResNet-152 + ISDA | **21.20%** | **5.67%** |
| ResNeXt-50, 32x4d [35] | 22.42% | 6.42% |
| ResNeXt-50, 32x4d + ISDA | **21.88%** | **6.23%** |

## 4.3 Comparison with Other Approaches

We compare ISDA with a number of competitive baselines described in Section 4.1, ranging from robust loss functions to semantic data augmentation algorithms based on generative models. The

Initial Restored Augmented Initial Restored Augmented

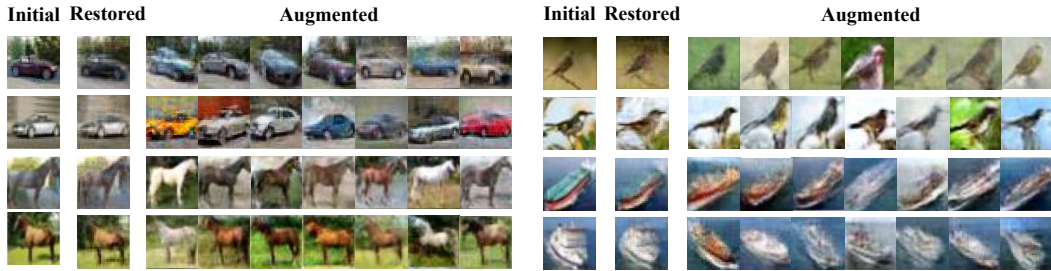

Figure 4: Visualization results of semantically augmented images.

results are summarized in Table 3, and the training curves are presented in Appendix D. One can observe that ISDA compares favorably with all the competitive baseline algorithms. With ResNet-110, the test errors of other robust loss functions are 6.38% and 27.85% on CIFAR-10 and CIFAR-100, respectively, while ISDA achieves 6.23% and 27.11%, respectively.

Among all GAN-based semantic augmentation methods, ACGAN gives the best performance, especially on CIFAR-10. However, these models generally suffer a performance reduction on CIFAR-100, which do not contain enough samples to learn a valid generator for each class. In contrast, ISDA shows consistent improvements on all the datasets. In addition, GAN-based methods require additional computation to train the generators, and introduce significant overhead to the training process. In comparison, ISDA not only leads to lower generalization error, but is simpler and more efficient.

### 4.4 Visualization Results

To demonstrate that our method is able to generate meaningful semantically augmented samples, we introduce an approach to map the augmented features back to the pixel space to explicitly show semantic changes of the images. Due to space limitations, we defer the detailed introduction of the mapping algorithm and present it in Appendix C.

Figure 4 shows the visualization results. The first and second columns represent the original images and reconstructed images without any augmentation. The rest columns present the augmented images by the proposed ISDA. It can be observed that ISDA is able to alter the semantics of images, e.g., backgrounds, visual angles, colors and type of cars, color of skins, which is not possible for traditional data augmentation techniques.

### 4.5 Ablation Study

To get a better understanding of the effectiveness of different components in ISDA, we conduct a series of ablation studies. In specific, several variants are considered: (1) *Identity matrix* means replacing the covariance matrix $\Sigma_c$ by the identity matrix. (2) *Diagonal matrix* means using only the diagonal elements of the covariance matrix $\Sigma_c$. (3) *Single*

Table 5: The ablation study for ISDA.

| Setting | CIFAR-10 | CIFAR-100 |
|---|---|---|
| Basic | 3.82±0.15% | 18.58±0.10% |
| Identity matrix | 3.63±0.12% | 18.53±0.02% |
| Diagonal matrix | 3.70±0.15% | 18.23±0.02% |
| Single covariance matrix | 3.67±0.07% | 18.29±0.13% |
| Constant $\lambda_0$ | 3.69±0.08% | 18.33±0.16% |
| ISDA | **3.58±0.15%** | **17.98±0.15%** |

*covariance matrix* means using a global covariance matrix computed from the features of all classes. (4) *Constant* $\lambda_0$ means using a constant $\lambda_0$ without setting it as a function of the training iterations.

Table 5 presents the ablation results. Adopting the identity matrix increases the test error by 0.05% on CIFAR-10 and nearly 0.56% on CIFAR-100. Using a single covariance matrix greatly degrades the generalization performance as well. The reason is likely to be that both of them fail to find proper directions in the deep feature space to perform meaningful semantic transformations. Adopting a diagonal matrix also hurts the performance as it does not consider correlations of features.

## 5 Conclusion

In this paper, we proposed an efficient implicit semantic data augmentation algorithm (ISDA) to complement existing data augmentation techniques. Different from existing approaches leveraging generative models to augment the training set with semantically transformed samples, our approach is considerably more efficient and easier to implement. In fact, we showed that ISDA can be formulated as a novel robust loss function, which is compatible with any deep network with the cross-entropy loss. Extensive results on several competitive image classification datasets demonstrate the effectiveness and efficiency of the proposed algorithm.

## Acknowledgments

Gao Huang is supported in part by Beijing Academy of Artificial Intelligence (BAAI) under grant BAAI2019QN0106 and Tencent AI Lab Rhino-Bird Focused Research Program under grant JR201914.

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
