[Supplementary Material]

# Supplementary Materials

## Appendix

## A  Implementation Details of ISDA.

**Dynamic estimation of covariance matrices.** During the training process using $\overline{\mathcal{L}}_\infty$, covariance matrices are estimated by:

$$\boldsymbol{\mu}_j^{(t)} = \frac{n_j^{(t-1)}\boldsymbol{\mu}_j^{(t-1)} + m_j^{(t)}\boldsymbol{\mu'}_j^{(t)}}{n_j^{(t-1)} + m_j^{(t)}}, \tag{1}$$

$$\Sigma_j^{(t)} = \frac{n_j^{(t-1)}\Sigma_j^{(t-1)} + m_j^{(t)}\Sigma'_j^{(t)}}{n_j^{(t-1)} + m_j^{(t)}} + \frac{n_j^{(t-1)}m_j^{(t)}(\boldsymbol{\mu}_j^{(t-1)} - \boldsymbol{\mu'}_j^{(t)})(\boldsymbol{\mu}_j^{(t-1)} - \boldsymbol{\mu'}_j^{(t)})^T}{(n_j^{(t-1)} + m_j^{(t)})^2}, \tag{2}$$

$$n_j^{(t)} = n_j^{(t-1)} + m_j^{(t)} \tag{3}$$

where $\boldsymbol{\mu}_j^{(t)}$ and $\Sigma_j^{(t)}$ are the estimates of average values and covariance matrices of the features of $j^{th}$ class at $t^{th}$ step. $\boldsymbol{\mu'}_j^{(t)}$ and $\Sigma'_j^{(t)}$ are the average values and covariance matrices of the features of $j^{th}$ class in $t^{th}$ mini-batch. $n_j^{(t)}$ denotes the total number of training samples belonging to $j^{th}$ class in all $t$ mini-batches, and $m_j^{(t)}$ denotes the number of training samples belonging to $j^{th}$ class only in $t^{th}$ mini-batch.

**Gradient computation.** In backward propagation, gradients of $\overline{\mathcal{L}}_\infty$ are given by:

$$\frac{\partial\overline{\mathcal{L}}_\infty}{\partial b_j} = \frac{\partial\overline{\mathcal{L}}_\infty}{\partial z_j} = \begin{cases} \frac{e^{z_{y_i}}}{\sum_{j=1}^C e^{z_j}} - 1, & j = y_i \\ \frac{e^{z_j}}{\sum_{j=1}^C e^{z_j}}, & j \neq y_i \end{cases}, \tag{4}$$

$$\frac{\partial\overline{\mathcal{L}}_\infty}{\partial \boldsymbol{w}_j^T} = \begin{cases} (\boldsymbol{a}_i + \sum_{n=1}^C[\lambda(\boldsymbol{w}_n^T - \boldsymbol{w}_{y_i}^T)\Sigma_i])\frac{\partial\overline{\mathcal{L}}_\infty}{\partial z_j}, & j = y_i \\ (\boldsymbol{a}_i + \lambda(\boldsymbol{w}_j^T - \boldsymbol{w}_{y_i}^T)\Sigma_i)\frac{\partial\overline{\mathcal{L}}_\infty}{\partial z_j}, & j \neq y_i \end{cases}, \tag{5}$$

$$\frac{\partial\overline{\mathcal{L}}_\infty}{\partial a_k} = \sum_{j=1}^C w_{jk}\frac{\partial\overline{\mathcal{L}}_\infty}{\partial z_j}, 1 \leq k \leq A, \tag{6}$$

where $w_{jk}$ denotes $k^{th}$ element of $\boldsymbol{w}_j$. $\partial\overline{\mathcal{L}}_\infty/\partial\boldsymbol{\Theta}$ can be obtained through the backward propagation algorithm using $\partial\overline{\mathcal{L}}_\infty/\partial\boldsymbol{a}$.

## B  Training Details

On CIFAR, we implement the ResNet, SE-ResNet, Wide-ResNet, ResNeXt, DenseNet and Pyramid-Net. The SGD optimization algorithm with a Nesterov momentum is applied to train all models. Specific hyper-parameters for training are presented in Table 1.

On ImageNet, we train ResNet and ResNeXt for 120 epochs using the same l2 weight decay and momentum as CIFAR, following [1]. The initial learning rate is set as 0.1 and divided by 10 every 30 epochs. The size of mini-batch is set as 256.

All baselines are implemented with the same training configurations mentioned above. Dropout rate is set as 0.3 for comparison if it is not applied in the basic model, following the instruction in [2]. For noise rate in disturb label, 0.05 is adopted in Wide-ResNet-28-10 on both CIFAR-10 and CIFAR-100

Table 1: Training configurations on CIFAR. '$l_r$' donates the learning rate.

| Network | Total Epochs | Batch Size | Weight Decay | Momentum | Initial $l_r$ | $l_r$ Schedule |
|---|---|---|---|---|---|---|
| ResNet | 160 | 128 | 1e-4 | 0.9 | 0.1 | Multiplied by 0.1 in $80^{th}$ and $120^{th}$ epoch. |
| SE-ResNet | 200 | 128 | 1e-4 | 0.9 | 0.1 | Multiplied by 0.1 in $80^{th}$, $120^{th}$ and $160^{th}$ epoch. |
| Wide-ResNet | 240 | 128 | 5e-4 | 0.9 | 0.1 | Multiplied by 0.2 in $60^{th}$, $120^{th}$, $160^{th}$ and $200^{th}$ epoch. |
| DenseNet-BC | 300 | 64 | 1e-4 | 0.9 | 0.1 | Multiplied by 0.1 in $150^{th}$, $200^{th}$ and $250^{th}$ epoch. |
| ResNeXt | 350 | 128 | 5e-4 | 0.9 | 0.05 | Multiplied by 0.1 in $150^{th}$, $225^{th}$ and $300^{th}$ epoch. |
| Shake Shake | 1800 | 64 | 1e-4 | 0.9 | 0.1 | Cosine learning rate. |
| PyramidNet | 1800 | 128 | 1e-4 | 0.9 | 0.1 | Cosine learning rate. |

Figure 1: Overview of the algorithm. We adopt a fixed generator $\mathcal{G}$ obtained by training a wasserstein gan to generate fake images for convolutional networks, and optimize the inputs of $\mathcal{G}$ in terms of the consistency in both the pixel space and the deep feature space.

datasets and ResNet-110 on CIFAR 10, while 0.1 is used for ResNet-110 on CIFAR 100. Focal Loss contains two hyper-parameters $\alpha$ and $\gamma$. Numerous combinations have been tested on the validation set and we ultimately choose $\alpha = 0.5$ and $\gamma = 1$ for all four experiments. For L$_q$ loss, although [3] states that $q = 0.7$ achieves the best performance in most conditions, we suggest that $q = 0.4$ is more suitable in our experiments, and therefore adopted. For center loss, we find its performance is largely affected by the learning rate of the center loss module, therefore its initial learning rate is set as 0.5 for the best generalization performance.

For generator-based augmentation methods, we apply the GANs structures introduced in [4, 5, 6, 7] to train the generators. For WGAN, a generator is trained for each class in CIFAR-10 dataset. For CGAN, ACGAN and infoGAN, a single model is simply required to generate images of all classes. A 100 dimension noise drawn from a standard normal distribution is adopted as input, generating images corresponding to their label. Specially, infoGAN takes additional input with two dimensions, which represent specific attributes of the whole training set. Synthetic images are involved with a fixed ratio in every mini-batch. Based on the experiments on the validation set, the proportion of generalized images is set as $1/6$.

## C   Reversing Convolutional Networks

To explicitly demonstrate the semantic changes generated by ISDA, we propose an algorithm to map deep features back to the pixel space. Some extra visualization results are shown in Figure 2.

An overview of the algorithm is presented in Figure 1. As there is no closed-form inverse function for convolutional networks like ResNet or DenseNet, the mapping algorithm acts in a similar way to [8] and [9], by fixing the model and adjusting inputs to find images corresponding to the given features. However, given that ISDA augments semantics of images in essence, we find it insignificant to directly optimize the inputs in the pixel space. Therefore, we add a fixed pre-trained generator $\mathcal{G}$, which is obtained through training a wasserstein GAN [4], to produce images for the classification model, and optimize the inputs of the generator instead. This approach makes it possible to effectively reconstruct images with augmented semantics.

The mapping algorithm can be divided into two steps:

**Step I.** Assume a random variable $z$ is normalized to $\hat{z}$ and input to $\mathcal{G}$, generating fake image $\mathcal{G}(\hat{z})$. $x_i$ is a real image sampled from the dataset (such as CIFAR). $\mathcal{G}(\hat{z})$ and $x_i$ are forwarded through a pre-trained convolutional network to obtain deep feature vectors $f(\mathcal{G}(\hat{z}))$ and $a_i$. The first step of

Figure 2: Extra visualization results.

the algorithm is to find the input noise variable $z_i$ corresponding to $x_i$, namely

$$z_i = \arg\min_{z} \|f(\mathcal{G}(\hat{z})) - a_i\|_2^2 + \eta\|\mathcal{G}(\hat{z}) - x_i\|_2^2, \; s.t. \; \hat{z} = \frac{z - \overline{z}}{std(z)}, \tag{7}$$

where $\overline{z}$ and $std(z)$ are the average value and the standard deviation of $z$, respectively. The consistency of both the pixel space and the deep feature space are considered in the loss function, and we introduce a hyper-parameter $\eta$ to adjust the relative importance of two objectives.

**Step II.** We augment $a_i$ with ISDA, forming $\tilde{a}_i$ and reconstructe it in the pixel space. Specifically, we search for $z_i'$ corresponding to $\tilde{a}_i$ in the deep feature space, with the start point $z_i$ found in Step I:

$$z_i' = \arg\min_{z'} \|f(\mathcal{G}(\hat{z}')) - \tilde{a}_i\|_2^2, \; s.t. \; \hat{z}' = \frac{z' - \overline{z'}}{std(z')}. \tag{8}$$

As the mean square error in the deep feature space is optimized to 0, $\mathcal{G}(\hat{z_i}')$ is supposed to represent the image corresponding to $\tilde{a}_i$.

The proposed algorithm is performed on a single batch. In practice, a ResNet-32 network is used as the convolutional network. We solve Eq. (7), (8) with a standard gradient descent (GD) algorithm of 10000 iterations. The initial learning rate is set as 10 and 1 for Step I and Step II respectively, and is divided by 10 every 2500 iterations. We apply a momentum of 0.9 and a l2 weight decay of 1e-4.

# D   Extra Experimental Results

(a) ResNet-110 on CIFAR-10

(b) ResNet-110 on CIFAR-100

Figure 3: Comparison with state-of-the-art image classification methods.

Curves of test errors of state-of-the-art methods and ISDA are presented in Figure 3. ISDA outperforms other methods consistently, and shows the best generalization performance in all situations. Notably, ISDA decreases test errors more evidently in CIFAR-100, which demonstrates that our method is more suitable for datasets with fewer samples. This observation is consistent with the results in the paper. In addition, among other methods, center loss shows competitive performance with ISDA on CIFAR-10, but it fails to significantly enhance the generalization in CIFAR-100.