[Reviews · NeurIPS 2019]

Reviewer 1



Originality: To the best of my knowledge, the paper main idea is novel, and I find it very interesting. Clarity: The paper is clear, well structured and well written. Quality: The initial motivation and the description of the method are sound. The derivation of the upper bound seems correct, and very useful. The experimental evaluation is well designed and provides a fair assessment of the method. One missing experiment/information in my opinion is the time cost for the estimation of the covariance matrices. Another interesting experiment would be to reduce the number of training samples and see if a smaller sample with a strong regularization by ISDA can achieve (close to) state-of-the-art results. Significance: The experimental results show that adding ISDA to state-of-the-art models improves significantly the results. It also outperforms the data augmentation state-of-the-art method. When applied alone, Dropout has a better performance, but the combination of the two regularizations gives the best results. Moreover, the ablation study shows the importance of using the full covariance matrix for the computation of the loss.

Reviewer 2



The idea of the paper is original to the best of my knowledge. Major references are being cited and the paper does a good job differentiating it from previously published papers. The paper is technically sound with correct derivations. Claims are supported by results and theoretical analysis. The assumption that the embeddings follow a Gaussian distribution seems strong to me, and the limitations of such assumption could be studied by the paper. Also, the generated images from the embeddings look interesting, but it would be nice to see results where the system fails. The paper is clearly written -- very easy to follow. A few details of the implementation are explained in the supplementary material. The most relevant aspect of the paper is perhaps its significance - the proposed approach has the potential to be used by many researchers in the field given its simplicity and effectiveness. After reading the rebuttal, I'm still happy with the paper. I think this paper should be accepted to NIPS.

Reviewer 3



This paper proposes a new image data-augmentation approach that adds class-dependent noise to the features (instead of input images). The idea of augmenting in the feature space is new and intuitive. The surrogate loss looks reasonably sound. The paper is well written. I have major concerns about the experimental results. In particular, the reported performance of the baselines looks much weaker than those in other papers. E.g., from [12] table.2, Wide-ResNet-28-10 on CIFAR10 has 3.9 top-1 error rate; while in the present paper, it's only 4.81 for the base model and 4.30 for the proposed approach, both of which are weaker than the base model in [12]. The same observation applies to other settings (different base models and datasets). The empirical comparison is mainly with other "robust losses", such as focal loss, etc which can be seen as "implicit" data augmentation. How about other popular data augmentation approaches, such as those proposed in [6, 12, etc] which perform "explicit" data augmentation? The paper claims the proposed approach brings little additional computational cost. Doesn't the computation of covariance matrices for each data instance (Line.6 in Alg.1) cause more computation? What's computation complexity, and how does it affect the run time empirically? Line.185: lost --> loss

[Author Response · NeurIPS 2019]

**General Concern: Complexity of ISDA.** Theoretically, the computational complexity of updating the covariance matrices at each iteration is $O(B \times D^2)$ (using the online update equations in Supplementary A), where $B$ is the batch size and $D$ is the feature space dimension. In comparison, a typical ConvNet with $L$ layers requires $O(B \times D^2 \times K^2 \times H \times W \times L)$ operations, where $K$ is the filer kernel size, and $H$ and $W$ are the height and width of feature maps, respectively. Consider ResNet-110 on CIFAR (C10 & C100) as an example, for which we have $K=3$, $H=W=8$ and $L=109$ (ignoring the last FC-layer), then the extra computation cost of ISDA is up to *four orders of magnitude less* than the total computation cost of the network.

Empirically, due to implementation issues, we observed a $5\%$ to $12\%$ increase in training time. Results are shown in Table 1. We will add these results and analysis in our revision.

Table 1: Additional computational cost.

| Dataset | Model | Real | Theoretical |
|---|---|---|---|
| C10+ | ResNet-110 | 4.65% | 0.002% |
| | DenseNet-100-12 | 5.50% | 0.024% |
| C100+ | ResNet-110 | 5.23% | 0.004% |
| | DenseNet-100-12 | 12.03% | 0.053% |
| ImageNet | ResNet-152 | 10.75% | 0.082% |

## Reviewer #1

**1. Experiments with Fewer Training Samples.** As suggested, we run ISDA on C100+ using ResNet-110, with a varying number of training samples. The results are shown in Table 2. It can be observed that ISDA achieves improvements consistently, and performance gain seems to be more notable with fewer samples. For example, with 20% of training samples, ISDA outperforms the baseline by 4.12%.

Table 2: Results with smaller datasets. ($r$: proportion of samples used for training.)

| $r$ | w/o ISDA | w/ ISDA |
|---|---|---|
| 100% | 28.67±0.44% | 27.57±0.46% |
| 80% | 30.99±0.33% | 30.29±0.03% |
| 60% | 34.89±0.76% | 33.47±0.35% |
| 40% | 41.82±0.86% | 39.71±0.38% |
| 20% | 56.28±0.80% | 52.16±0.45% |

## Reviewer #2

Figure 1: Failures.

**1. Failure Cases.** We collect some cases when ISDA fails to produce meaningful semantic transformations, as shown in Fig. 1. Failures usually occur when an input image shows great semantic differences from typical images in its class. For example, the first image in Fig. 1 shows only the head of a bird, while most images show the entire body. A plausible explanation is that the semantic directions for these images are not well captured by the covariance matrix which is dominated by the majority of typical samples.

**2. Training Curves on CIFAR.** Thanks for the suggestion. We will update our paper with training curves. Notably, ISDA consistently achieves a slightly higher training error but lower test error, indicating its regularization effect.

**3. Tightness of the Upper Bound $\overline{\mathcal{L}}_\infty$.** The upper bound follows from the Jensen's inequality $\mathrm{E}[logX] \leq log\mathrm{E}[X]$, and the equation holds when $\lambda\Sigma_i \to 0$. To check the tightness of $\overline{\mathcal{L}}_\infty$ in practice, we empirically calculate $\mathcal{L}_\infty$ and $\overline{\mathcal{L}}_\infty$ over the training iterations, where $\mathcal{L}_\infty$ is estimated using Monte-Carlo sampling with sample size 1000, shown in Fig. 2. We can observe that $\overline{\mathcal{L}}_\infty$ gives a very tight upper bound.

Figure 2: Values of $\mathcal{L}_\infty$ and $\overline{\mathcal{L}}_\infty$.

**4. Gaussian Assumption.** Indeed, the Gaussian assumption seems to be strong. But as we discussed in the introduction, formulating the true distribution requires to find all possible semantic directions, which is practically intractable. Our algorithm achieves a nice tradeoff between tractability and accuracy by making this assumption. Please refer to the 3rd paragraph in our Introduction and the first paragraph in Section 3.1 for detailed discussion.

## Reviewer #3

Table 3: Different configurations

| | ISDA | AutoAugment |
|---|---|---|
| Total Epochs | 160 | 200 |
| Weight Decay | 1e-4 | 5e-4 |
| Cosine Learning Rate | × | ✓ |

**1. Weak Baseline Results.** Thanks for pointing out this issue. We have carefully checked our code and results during the rebuttal period, and find that some of our reproduced results on CIFAR (mostly for Wide-ResNet) are indeed worse than that reported in existing work. The reason is that we reproduced all these results by ourselves (in order to give a clean comparison), and we used the hyperparameters for ResNets to train Wide-ResNets, which tend to give inferior results (Wide-ResNets used improved training techniques). These differences are listed in Table 3. In addition, most existing work use all 50,000 samples for training and usually report the best results over iterations, while we held out 5,000 from the training set for validation.

Although with the above issues, we argue that our results are still valid, because (1) most of the baseline results are not affected and are competitive; and (2) even for affected cases, our comparison is fair because our ISDA used the same hyperparameters as the baselines.

After fixing the hyperparameter settings, we successfully reproduced stronger baselines. The new results are shown in the first two rows in Table 4, and ISDA still leads to better results. We will update the paper with these results.

**2. Comparisons with Explicit Augmentation.** Thanks for the suggestion. We have experimented with several recently proposed explicit data augmentation techniques, i.e., Cutout(CT), Random Erasing (RE) and AutoAugment (AA). From Table 4, it is clear that ISDA can still consistently improve their performance. In fact, our algorithm performs data augmentation in the *feature* space, and it is *complementary* to those explicit augmentation techniques in the *input* space. This can also be validated by our results on CIFAR w/ and w/o data augmentation shown in Table 1 in the paper.

Table 4: Comparisons with explicit augmentation methods.

| Method | C10+ | C100+ |
|---|---|---|
| WRN-28 | 3.82±0.15% | 18.58±0.10% |
| WRN-28 + ISDA | **3.58±0.15%** | **17.98±0.15%** |
| WRN-28 + CT | 2.99±0.06% | 18.05±0.25% |
| WRN-28 + CT+ISDA | **2.83±0.04%** | **17.02±0.11%** |
| WRN-28 + RE | 3.10±0.11% | 17.98±0.28% |
| WRN-28 + RE + ISDA | **2.95±0.09%** | **17.03±0.24%** |
| WRN-28 + AA | 2.65±0.07% | 16.63±0.17% |
| WRN-28 + AA + ISDA | **2.56±0.01%** | **15.38±0.11%** |

[Meta-Review · NeurIPS 2019]

Good paper, Accept. It is important that the authors include the new results in the revised version.